# Detection of Pancreatic Ductal Adenocarcinoma by Ex Vivo Magnetic Levitation of Plasma Protein-Coated Nanoparticles

**DOI:** 10.3390/cancers13205155

**Published:** 2021-10-14

**Authors:** Luca Digiacomo, Erica Quagliarini, Vincenzo La Vaccara, Alessandro Coppola, Roberto Coppola, Damiano Caputo, Heinz Amenitsch, Barbara Sartori, Giulio Caracciolo, Daniela Pozzi

**Affiliations:** 1NanoDelivery Lab, Department of Molecular Medicine, Sapienza University of Rome, Viale Regina Elena 291, 00161 Rome, Italy; luca.digiacomo@uniroma1.it (L.D.); giulio.caracciolo@uniroma1.it (G.C.); 2Department of Chemistry, Sapienza University of Rome, P.le Aldo Moro 5, 00185 Rome, Italy; erica.quagliarini@uniroma1.it; 3Department of Surgery, University Campus Bio-Medico di Roma, Via Alvaro del Portillo 200, 00128 Rome, Italy; v.lavaccara@unicampus.it (V.L.V.); a.coppola@unicampus.it (A.C.); r.coppola@unicampus.it (R.C.); 4Institute of Inorganic Chemistry, Graz University of Technology, Stremayrgasse 9/IV, 8010 Graz, Austria; amenitsch@tugraz.at (H.A.); barbara.sartori@elettra.eu (B.S.)

**Keywords:** pancreatic ductal adenocarcinoma, magnetic levitation, protein corona, graphene oxide, early cancer diagnosis

## Abstract

**Simple Summary:**

Pancreatic Ductal Adeno Carcinoma (PDAC) is a highly lethal disease, for which mortality closely parallels incidence. The poor prognosis of PDAC is mainly due to cancer’s biological behavior and its advanced stage at the moment of the diagnosis. Despite strong efforts in the scientific community, reaching an effective early diagnosis is still a big challenge. Our research aims to develop highly sensitive and specific technologies for the early diagnosis of PDAC using magnetic levitation (MagLev) of nanoparticles coated by the personalized protein corona, i.e., the protein layer of plasma proteins that bind to nanomaterials exposed to patients’ bodily fluids. MagLev does not rely on critical experimental steps (e.g., isolating plasma proteins from nanoparticles) and thus overcomes limitations that propagate biases, hinder reproducibility, and typically impair clinical translation of medical technologies.

**Abstract:**

Pancreatic Ductal Adeno Carcinoma (PDAC) is one of the most lethal malignancies worldwide, and the development of sensitive and specific technologies for its early diagnosis is vital to reduce morbidity and mortality rates. In this proof-of-concept study, we demonstrate the diagnostic ability of magnetic levitation (MagLev) to detect PDAC by using levitation of graphene oxide (GO) nanoparticles (NPs) decorated by a biomolecular corona of human plasma proteins collected from PDAC and non-oncological patients (NOP). Levitation profiles of corona-coated GO NPs injected in a MagLev device filled with a paramagnetic solution of dysprosium(III) nitrate hydrate in water enables to distinguish PDAC patients from NOP with 80% specificity, 100% sensitivity, and global classification accuracy of 90%. Our findings indicate that Maglev could be a robust and instrumental tool for the early detection of PDAC and other cancers.

## 1. Introduction

Pancreatic Ductal Adeno Carcinoma (PDAC) is a highly lethal [1], worldwide increasing [2] disease predicted to be the second cause of cancer-related deaths in the next decade [3]. Its poor prognosis is mainly due to cancer’s biological behavior and its advanced stage at the moment of the diagnosis [4]. Indeed, PDAC is often asymptomatic in its early stages or presents non-specific symptoms [5]. Moreover, despite the availability of improved radiological exams (e.g., multidetector CT scan, MRI, etc.), to date, there is a lack of effective diagnostic modalities to early detect pancreatic cancer, even in the presence of pancreatic masses [6]. To date, five-year survival rates for PDAC are about 10% and this percentage may increase up to 20% if a radical surgical resection is performed. However, in most cases, surgery is precluded for patients, because of the presence of distant metastases or invasion of vascular structures at the moment of the diagnosis [7]. For these reasons, the early diagnosis of PDAC would represent an important gap to fill in the treatment of this disease, on the other hand, complementary tests could aid in the diagnosis and play a role in surveillance intervals. Despite strong efforts in the scientific community, reaching an effective early diagnosis is still a big challenge. Indeed, the only PDAC biomarker approved by the US Food and Drug Administration (FDA) is carbohydrate antigen (CA) 19.9, but it exhibits non-adequate sensitivity (60–70%) and specificity (70–85%) values [8]. Thus, researchers have been aiming at identifying new sensitive PDAC biomarkers in the last few years, thanks to the improvement of molecular technologies such as genetic sequencing, transcriptomic expression profiling, metabolomics, proteomics [9,10,11], and glycomics [12]. Proteomic approaches are among the most powerful tools for potential biomarker identification, especially via mass spectrometry [11]. However, the sensitivity and specificity of such approaches for early PDAC detection are too low, mainly due to the low levels of protein biomarkers in human plasma (HP) with the result that many PDAC patients go undiagnosed [7,8]. Recently, glycomics profiling approaches have also gained importance since glycans represent important cancer hallmarks [12]. Indeed, Ca 19.9 levels increase in gastrointestinal cancer because of a cancer-associated aberrant glycosylation [13].

On the other hand, the employment of nanomaterials in biomedical research outlined promising perspectives in the field. Indeed, in recent years, a new paradigm is emerging that shows understanding how nanoparticles (NPs) interact with bodily fluids of cancer patients may lead to major advances in early cancer detection. This arises from the concept of protein corona (PC). According to current understanding [14], PC is defined as a cloud of proteins that adsorb on the NP’s surface after exposure to blood, plasma, or other biological fluids. As the composition of the corona depends on both the nanomaterial and the biological source [15,16], anomalies in the protein composition of the corona may be detected and associated with specific malignancies [17,18]. In other words, proteins adsorb on nanomaterials’ surfaces according to their chemical and physical affinities, and thus nanomaterials act as “nano-concentrators”. Therefore, it is in principle possible to detect potential cancer biomarkers, which have very low expression levels in plasma but relatively high affinity with the employed NP [19]. A turning point in this field was achieved when Mahmoudi and coworkers introduced the concept of ‘‘personalized’’ and ‘‘disease-specific’’ protein corona [20,21] and they demonstrated that exposing NPs to HP obtained from cancer patients and non-oncological subjects caused considerable differences in the corona profile of NPs. Of fundamental importance to develop effective diagnostic tools by PC technology is the choice of techniques to be used to evaluate the PC patterns, as well as the nanomaterials to be employed as “nano-concentrators”. According to the World Health Organization (WHO), the experimental procedures for cancer screening and detection must satisfy the ASSURED (Affordable, Sensitive, Specific, User-friendly, Rapid and robust, Equipment-free and Deliverable to end-users) criteria.

In this work, we present the Magnetic Levitation (MagLev) technique as a tool for PDAC detection, discuss its potential and the possible perspectives of optimization towards an effective use in the clinical practice. MagLev is a reliable, portable, and simple technique that separates objects according to their densities [22]. Although originally MagLev was mainly used for density-based analyses of bulk materials and microparticles [23,24], its employment has been recently extended to study and separate biomolecules, e.g., plasma proteins [25], as proteins have different densities because of their different molecular weights, shapes, and volumes. Furthermore, since different diseases specifically alter the protein levels in plasma, it has been hypothesized that the levitation progress and patterns of plasma proteins may contain some information on the health spectrum [22] and it has been shown that MagLev technology can be successfully used to study the PC of NPs [26]. Consequently, now MagLev represents a promising technique in the context of precision medicine and personalized PC. Unlike most of the existing procedures for PC analysis, MagLev allows an indirect characterization of the personalized PC, i.e., without isolating plasma proteins from NPs, and thus overcoming numerous and laborious steps of isolation and sample preparation that are typically performed by other techniques, including liquid chromatography tandem mass spectrometry (LC-MS/MS) or sodium dodecyl sulfate-polyacrylamide gel electrophoresis (SDS-PAGE). In this framework, for PC-based technology, a crucial factor is represented by the choice of the nanomaterial. In this work, we used Graphene Oxide (GO) as a nanoplatform. Due to its high specific surface area and the presence of carboxylic and epoxydic groups on its surface, GO exhibits high protein binding [27] and thus it represents an ideal candidate to differentiate HP samples derived from different donors [28]. Particularly, the PC formed on GO has been demonstrated to contain valuable information for early detection of PDAC [29,30].

Here, we present as a proof of concept the possibility to evaluate differences among the levitation patterns of GO-PC complexes obtained from non-oncological patients (NOP) and PDAC patients, by a straightforward experimental procedure followed by an image processing tool. After a thorough preliminary characterization of the experimental setup, we were able to investigate both the dynamic behavior of the samples and their equilibrium states and found that they depend on the specific class of donors. Indeed, by coupling the information arising from the levitation profiles of 20 plasma samples (10 from NOP and 10 from PDAC patients), a linear discriminant analysis resulted in high sensitivity and specificity values, and an overall classification ability of 90%. We, therefore, believe that this study shall represent a starting point towards the effective development of fast, portable, and robust devices for the early diagnosis of PDAC and other cancers.

## 2. Materials and Methods

### 2.1. Preparation of Graphene Oxide

Graphene Oxide (GO) solution was purchased from Graphenea (San Sebastián, Spain), subjected to sonication (Vibra cell sonicator VC505, Sonics and Materials, Newton, CT, USA) and centrifugation (Hermle Z 216 MK, Hermle Labortechnik, Wehingen, Germany), to obtain GO sheets with a size of about 400 nm. Finally, GO concentration was measured by UV-Vis spectrophotometry, and—in this work—it was kept to 0.25 mg/mL. Further details of GO sizing, centrifugation and characterization can be found elsewhere [28].

### 2.2. Human Plasma

For preliminary experiments, commercially available human plasma (HP) was purchased from Sigma−Aldrich, Inc. (Merk KGaA, Darmstadt, Germany). Lyophilized HP was dissolved in water according to the manufacturer’s instructions, then clarified by centrifugation, and finally stored at −20 °C.

### 2.3. Patients’ Enrolment and Blood Sample Collection

This work was approved by the Ethical Committee of the University Campus Bio-Medico di Roma approved (Prot. 10/12 ComEt CBM). The study group comprised 10 non-oncological subjects (healthy control) and 10 PDAC patients (PDAC group). A validation cohort comprised 5 non-oncological subjects (healthy control) and 5 PDAC patients (PDAC group). All the participants were cytohistologically diagnosed and proven to be eligible. The inclusion criteria for the study were age > 18 years; adequate renal function (creatinine < 1.5 mg/dL, blood urea nitrogen < 1.5 times the upper limit). Previous personal medical history negative for neoplasticity; renal or liver disease or blood disorders; no previous chemotherapy or radiotherapy; absence of uncontrolled infections; and written, informed consent. Demographic characteristics of NOP and PDAC patients are reported in Appendix A, while comorbidities are reported in Appendix A. Demographic characteristics of NOP and PDAC patients of the validation cohort are reported in Appendix A, and comorbidities are reported in Appendix A. The diagnosis of PDAC was confirmed by radical surgery. Blood samples were collected by venipuncture and stored in TM BD P100 Blood Collection System (Franklin Lakes, NJ, USA) comprising test tubes with K2EDTA and a protease inhibitor solution. Plasma was obtained by centrifugation and stored according to standard procedures [31].

### 2.4. Preparation of GO-HP Samples

GO-protein complexes were obtained by incubating 50 μL of GO solutions (0.25 mg/mL) with 20 μL of HP from healthy donors and PDAC patients at 37 °C for 1 h. 30 μL of distilled water was added to reach a total sample volume of 100 μL.

### 2.5. MagLev Device

The Maglev instrument consisted of two N42-grade neodymium (NdFeB) cubic magnets (25.4 mm length, 25.4 mm width, and 50.8 mm height, purchased from Magnet4less) with North poles facing each other at a fixed separation distance of 28 mm (Figure 1a,b). A 4 mL plastic cuvette was properly cut to 25 mm and used as a sample container. Dysprosium (III) nitrate hydrate (from Sigma−Aldrich, Inc. Merk KGaA, Darmstadt, Germany) was dissolved in water at different concentrations and the resulting solutions were used as paramagnetic media for MagLev experiments.

After filling the MagLev cuvette with the paramagnetic medium at the desired concentration, each of the GO-HP samples was injected at the bottom of the cuvette. The injection was performed maintaining a vertical orientation of the pipet tip. Due to density differences, under all the explored experimental conditions (e.g., concentration of samples and paramagnetic solutions, volumes, amount of plasma proteins) the sample lifted towards the upper part of the cuvette and distributed homogeneously near the liquid surface. Then, the cuvette was inserted between the magnets of the MagLev device (Appendix A).

The Maglev device has a capacity to levitate a diamagnetic object in a paramagnetic solution when the induced magnetic force ${\vec{F}}_{mag}$ is strong enough to cancel out the gravitational force ${\vec{F}}_{g}$ [32,33].
(1)$${\vec{F}}_{mag}+{\vec{F}}_{g}=0$$
where ${\vec{F}}_{mag}$ depends on the magnetic susceptibility of the paramagnetic medium ($\chi_{m}$), the magnetic susceptibility ($\chi_{s}$) and the volume (V) of the diamagnetic object, the magnetic field ($\vec{B}$), and the magnetic permeability of free space ($\mu_{0}$), as follows
(2)$${\vec{F}}_{mag}=\frac{\chi_{s}-\chi_{m}}{\mu_{0}}V\left(\vec{B}\cdot \vec{\nabla }\right)\vec{B}$$
and ${\vec{F}}_{g}$ is the buoyancy-corrected gravitational force, i.e.,
(3)$${\vec{F}}_{g}=\left(\rho_{s}-\rho_{m}\right)V\vec{g}$$
where $\rho_{s}$ and $\rho_{m}$ represent the sample density and medium density, respectively, and $\vec{g}$ is the gravity acceleration. At the equilibrium (Equation (1)), taking into account the expression of the magnetic force (Equation (2)), the gravitational force (Equation (3)), and the geometry of the magnetic setup [25] the diamagnetic sample reaches a steady height h, which reads
(4)$$h=\frac{d}{2}+\frac{\left(\rho_{s}-\rho_{m}\right)g\mu_{0}d^{2}}{\left(\chi_{s}-\chi_{m}\right)4B^{2}}$$
where d is the distance between the permanent magnets. Of note, objects can levitate in the MagLev platform only if Equation (1) can be satisfied. Otherwise, they precipitate at the bottom of the cuvette. In this work both steady components and precipitating populations of samples were studied by acquiring image series of MagLev patterns (at a controlled temperature of 23 °C) and processing them by custom Matlab (Mathwork, Portola Valley, CA, USA) scripts.

## 3. Results

MagLev is a fast, cheap, and portable technique, whose configuration consists of two permanent magnets with the like-poles facing each other at a fixed (but tunable) distance d (Figure 1a). Diamagnetic objects embedded in a paramagnetic solution and placed in a cuvette within the magnets move by the effect of a magnetic field $\vec{B}$. The intensity of the magnetic field is very high (about 0.5 T) next to surfaces of the magnets and steeply reaches 0 at *d*/2. A schematic representation of the experimental setup is reported in Figure 1b. As predicted by theoretical considerations (which are outlined in the Materials and Methods Section), manifold factors affect the levitation patterns of diamagnetic objects, including density of sample and medium, magnetic susceptibilities, distance of the magnets, and field intensity. Then, as a first step of our study, we explored the role of the paramagnetic medium concentration, by fixing all the other parameters. To this end, we dissolved proper amounts of dysprosium(III) nitrate hydrate in water to obtain paramagnetic solutions over two orders of magnitude of concentration, ranging from 1 mg/mL to 120 mg/mL. Commercially available human plasma (HP) was diluted 1:20 and a fixed sample volume (i.e., 100 μL) was injected into the MagLev tube.

Representative outcomes of MagLev responses are shown in Figure 1c, d. For low values of dysprosium concentration, HP fully precipitated in about 5 min. On the other hand, within a concentration range of 25–60 mg/mL, HP exhibited a separated pattern. Indeed, the precipitating fraction reached the bottom of the cuvette in about 15 min, whereas a stable population levitated in the upper part of the tube. Finally, for concentrations higher than 60 mg/mL, HP totally levitated up to 50–60 min (Figure 1e). Beyond this timescale, a slight precipitating behavior occurred. Aiming to maximize the available information from the MagLev systems, we chose to analyze deeper the “separation condition”, as decoupling the behavior of the levitating fraction and the precipitating fraction may give deeper insight for diagnostic purposes. In this regard, image time-series of MagLev patterns were acquired and processed to quantify both the steady states and the sample’s dynamics.

A representative example of the developed image processing tool is shown in Figure 2. A generic frame of the image time-series (Figure 2a) can be regarded as an intensity function that is sampled over a two-dimensional matrix (Figure 2b). The vertical projection that is evaluated over a region of interest represents the MagLev one-dimensional profile (Figure 2c). When two populations are detected, their relative abundances can be quantified by studying the intensity signal, i.e., by determining locations, amplitude, and widths of the corresponding peaks. Interestingly, although this optical evaluation cannot be used for absolute quantification of protein contents, the low-to-high density population ratio agreed with the results obtained from bicinchoninic acid assay (details are provided in Appendix A), which represents a standard procedure to measure the total protein concentration in solution [34].

Through these preliminary steps, we determined the optimal concentration of the paramagnetic medium to achieve a complete separation of protein samples and implemented a computational tool to quantify the patterns. As a next step, we proceeded to investigate the ability of MagLev to distinguish PDAC patients from NOP. To this end, we selected 10 NOP and 10 PDAC patients meeting the inclusion criteria reported in the Materials and Methods section. In the selected cohort we measured systemic inflammatory response biomarkers (SIRBs) such as white blood count (WBC), neutrophils to lymphocytes ratio (NLR), derived-NLR (d-NLR), and platelets to lymphocytes ratio (PLR). SIRBs have attracted considerable attention for the diagnosis and prognosis of different types of tumors, including PDAC. As Appendix A clearly shows, no SIRB value, alone or in combination with the others, was clearly predictive of PDAC. For MagLev validation, we exposed GO NPs to HP collected from NOP and PDAC patients (a thorough characterization of GO NPs and GO-HP complexes is reported in Appendix A). After 1-hour incubation at 37 °C, each of the samples was injected in a dysprosium solution (concentration = 50 mg/mL) and the system was inserted in the MagLev device. Two representative examples of levitation patterns are shown in Figure 3 for (a) NOP and (b) PDAC, respectively. Both classes of samples were exhibited.

A remarkable precipitating component, which coexisted with a slight –but detectable- levitating population at a height of about 16 mm. Interestingly, the distributions of the intensity profiles at the last frames (Figure 3c) revealed slightly different trends for NOP and PDAC. Indeed, the amplitude of the main peak, which corresponds to the precipitating component, was slightly larger for NOP than for PDAC, whereas this behavior was inverted and less prominent for the secondary peak, which corresponds to the levitating population. Distributions of integral areas are reported in Figure 3d,e, along with the *p*-values from Student’s *t*-test. Furthermore, a study of the entire image time series revealed that the precipitating population moved with a faster dynamic for NOP than for PDAC (Figure 3f). In this specific configuration (i.e., nature and concentration of the paramagnetic solution, geometry of the MagLev device, density of levitating objects, etc.) the starting position (Figure 3g) and precipitation speed (Figure 3h) of coronated GO NPs could be identified as MagLev fingerprints for PDAC. Due to all this evidence, we suggest that NOP samples contain a larger precipitating population, which therefore starts its MagLev dynamics at a lower height and reaches the bottom of the cuvette faster than the PDAC counterparts. This behavior represents an average trend, but intra-class variability generally resulted in broad distributions of the measured parameters (Figure 3d,e,g,h). Thus, we coupled all the available information to compute a classification analysis and finally evaluate the specificity and sensitivity of a MagLev-based test for PDAC detection.

Among the possible six couplings of four parameters, the best classification was obtained by linear discriminant analysis (LDA) over the multivariate distributions of the pattern’s starting position and the relative amount of the levitating population, i.e., the levitating component. Figure 4a reports the scatter plot of the quantities for NOP (blue) and PDAC (orange) samples. Ellipses depict the extensions of the two-variable distributions, and the black solid line represents the delimiter item from LDA. All the measurements were performed in triplicates and the experimental errors for each sample are included in Figure 4a as error bars, along with an inter-user reproducibility test (Appendix A and Appendix A). According to this statistical analysis, the parameter space was subdivided into a bottom-left region corresponding to the control patterns and a top-right region corresponding to the PDAC ones. As Figure 4a clearly shows, only two PDAC samples were misclassified, thus resulting in a test with a global classification accuracy of 90% (specificity = 100% and sensitivity = 80%).

To validate the aforementioned classification by MagLev fingerprints, a blind validation test was performed on a cohort of 5 NOP and 5 PDAC samples. As shown in Figure 4b, only one NOP sample was misclassified by the test, which thus reached a global accuracy value of 90%.

## 4. Discussion

Many more people are living with PDAC worldwide as the result of an aging population, unhealthy lifestyles, and unfavorable social conditions [35]. This is generating a huge burden for citizens, cancer patients, survivors, and their families, and for health systems and society at large. Even today, less than 10% of patients survive five years after diagnosis. To reverse these dramatic statistics, early detection of PDAC, when the tumor is small and distant metastases are absent, may represent an important gap to fill in the treatment and management of this disease.

Furthermore, it is known that even the most performing available diagnostic technologies (CT scan, MRI, EUS +/− FNA, etc.) are burdened by diagnostic failures due to the characteristics of the primary tumor (e.g., isodense tumors regarding the surrounding parenchyma) and of the patients (e.g., presence of metal implants) or to the skills of the operator and by toxicity (e.g., nephrotoxicity, reactions to the contrast medium, radiation exposure) [6].

This pushes the scientific community to develop new technologies for the early diagnosis of this lethal neoplasm. In recent years, joint efforts by oncologists and material scientists have focused on studying the interaction between nanomaterials and blood samples from cancer patients. Some of us have shown that nanomaterials of various kinds exposed to human blood samples are coated with a personalized layer of plasma proteins called “personalized” PC (PPC) [15,20,21,36]: everyone has his own PC and, in the event of cancer, recognizable alterations in composition occur. In a series of recent papers, we characterized the PPC of PDAC patients by proteomics techniques such as SDS-PAGE and LC-MS/MS demonstrating that it is distinguishable from that of NOP and from patients with other types of cancer [28,29]. These studies led to the development of sensor arrays of nanoparticles [28] and nanoparticle-enabled blood (NEB) tests [37] for early cancer detection with higher sensitivity and specificity than those of Ca-19.9, the only PDAC biomarker approved by the US FDA so far. Despite these promising outcomes, isolating plasma proteins from NPs by SDS-PAGE and LC-MS/MS does not meet the ASSURED criteria for early cancer detection stated by WHO. Direct methods are not user-friendly as they depend on numerous and laborious steps (e.g., protein isolation, preparation for analysis, and bioinformatic identification) and are also expensive and time-consuming. Moreover, isolating proteins from nanomaterials is largely user-dependent and may lead to high variability in experimental outcomes thus impairing consistency among laboratories. To surpass current technological paradigms, indirect characterization of the personalized protein corona (i.e., without isolating plasma proteins from NPs) is a key step. In this regard, MagLev technology offers outstanding advantages, due to its intrinsic properties. Indeed, the MagLev device consists of two strong permanent magnets that flank a cuvette filled with a solution of a paramagnetic medium. When diamagnetic objects (e.g., plasma proteins or NP-PC systems) are simply injected into the test tube of a MagLev device, they levitate and equilibrate at different heights depending on the intensity of the magnetic field gradient, exposure time, and, most importantly, on the object density. As a result, the output of a MagLev measurement is a series of optical images, which can be processed to study the behavior of the diamagnetic samples and to quantify the levitation patterns by physical parameters.

The latest studies have demonstrated that MagLev is a fast, user-friendly, non-destructive, and powerful tool to separate substances with different densities. The versatility of the MagLev platform is ascribable to the large number of tunable variables, which directly and indirectly affect the levitation patterns and therefore should be adjusted according to the desired applications. One of the most impacting factors in the MagLev setup is the choice of the paramagnetic solution and its concentration, as the concentration affects both the solution’s density and magnetic susceptibility. For this reason, it was necessary to evaluate the levitation response of HP at different concentrations of dysprosium(III) nitrate hydrate, which to the authors’ knowledge has never been used before in magnetic levitation experiments. Interestingly, three different behaviors were found. HP fully precipitated in 5 min for low dysprosium concentration, totally levitated up to 50 min for high dysprosium concentration and the coexistence of a stable levitating population with a precipitating one was found at an intermediate dysprosium concentration (Figure 1). Thus, we chose the latter condition (concentration 50 mg/mL) as the optimal one, as two distinct profiles enhance the available information to perform further classification analysis for diagnostic purposes. Indeed, with that configuration, we demonstrated that the behavior of corona-coated NPs in a MagLev device contains “fingerprints of PDAC”. By employing GO as a nanoplatform for protein corona formation, our study over a dataset of 10 NOP and 10 PDAC samples clearly indicated that differences between the two classes can be found in the relative amount of low-density and high-density populations and in the kinetic parameters of the precipitating component (Figure 3). According to the study of both the steady states and the time evolution of the MagLev patterns, the low-density population for PDAC samples was more abundant than that of one of the NOP counterparts. Consequently, the starting position of MagLev profiles for PDAC was located at higher values and the high-density components precipitated more slowly. To interpret results properly we evaluated total protein content and albumin. Results reported in Appendix A demonstrate that significantly less albumin was present in PDAC patients. Such higher protein concentration may result in faster precipitation thus accounting, at least in part, for the lower starting position observed for NOP (Figure 3g). Considering the whole distributions for all the available quantities, and by coupling the low-density component with the starting position of the precipitating population, we were able to correctly classify 90% of the samples, with 100% specificity and 80% sensitivity (Figure 4). When Ca-19.9 and CEA levels were plotted against distance from the cutoff, no clear correlation was found (Appendix A). While a straightforward interpretation is beyond the scope of this investigation, we speculate that protein alteration inducing distinct MagLev patterns in PDAC patients may be not related to the biological variation of blood PDAC biomarkers.

This study is not without limitations such as the small size of the population and the poor representation in the sample of early-stage cancers. Unfortunately, the low rate of stage I and II cancers represents the current and real picture of the epidemiology of pancreatic cancer which is mostly diagnosed at an already advanced stage. Despite the small number of investigated samples in this proof-of-concept work, this promising outcome may represent a starting point for further studies. Indeed, this approach would benefit from the existence of several tunable parameters (e.g., intensity of magnetic field gradient, exposure time, NP size) and will not suffer the aforementioned limitations arising from direct measurements of corona composition. The possibility to exploit the “MagLev fingerprint” for early cancer detection has never been explored so far and will represent a truly new paradigm for early diagnosis of PDAC. Furthermore, as MagLev is a non-destructive test, isolating coronas of PDAC patients and NOP can be easily extracted and further analyzed by LC-MS/MS. The application of these new technologies could be disruptive in the clinical setting to screen people at risk for pancreatic cancer, such as people with obesity or diabetes, and to play a role as complementary tests in surveillance intervals. Ongoing and future studies could also help better understand the staging of this tumor and the response of patients to neoadjuvant therapies.

## 5. Conclusions

In summary, we demonstrated that ex vivo levitation profiles of corona-coated NPs contain useful information about the health spectrum of PDAC patients in paramagnetic solutions. If further validated in larger-cohort investigations, Maglev may be helpful in the fast and robust detection of PDAC with potential impact on clinical practice. To extend the detection capacity of Maglev, the sensitivity, specificity, and prediction accuracy of the system needs to be tested by systematic changes in the numerous factors affecting the particle levitation profiles such as nature and concentration of the paramagnetic solution, NP type, exposure time to plasma samples.

## Figures and Tables

**Figure 1 cancers-13-05155-f001:**
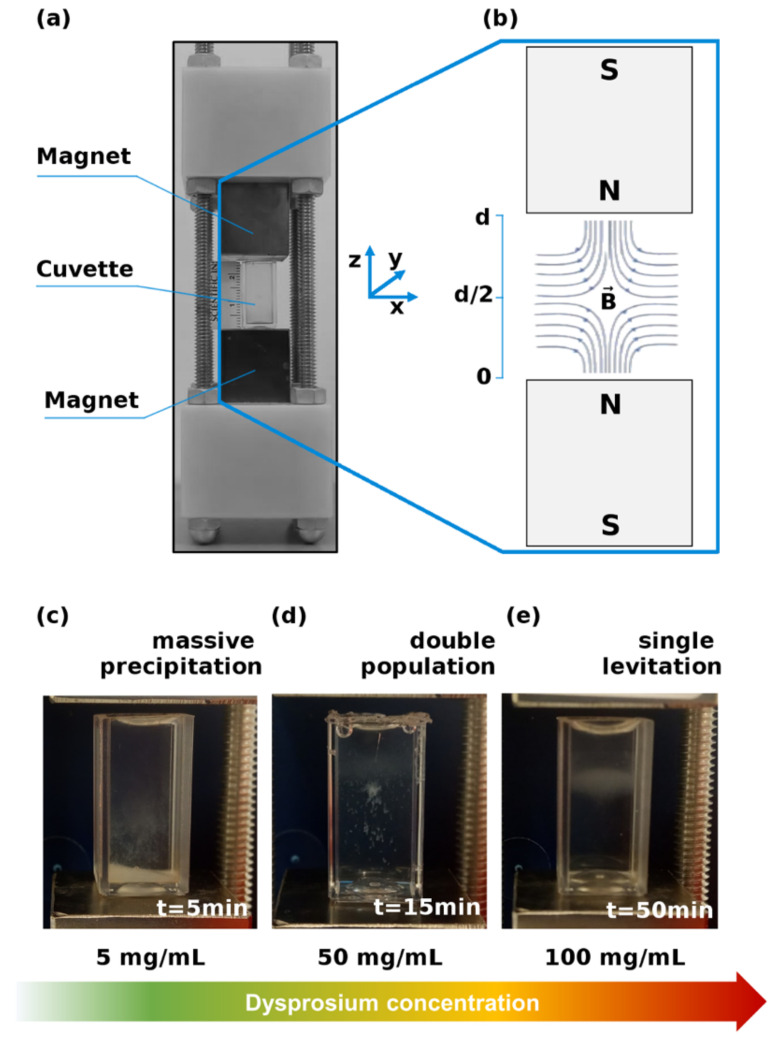
(**a**) Photograph and (**b**) scheme of the MagLev platform. The magnetic levitation patterns of HP in dysprosium nitrate depended on the concentration of the paramagnetic solution. (**c**) At low concentrations, HP precipitated in 5 min. (**d**) At intermediate concentrations, a stable levitating population coexisted with a precipitating one, which reached the bottom in about 15 min. (**e**) At high concentrations, HP levitated in a larger time scale (up to about 50 min).

**Figure 2 cancers-13-05155-f002:**
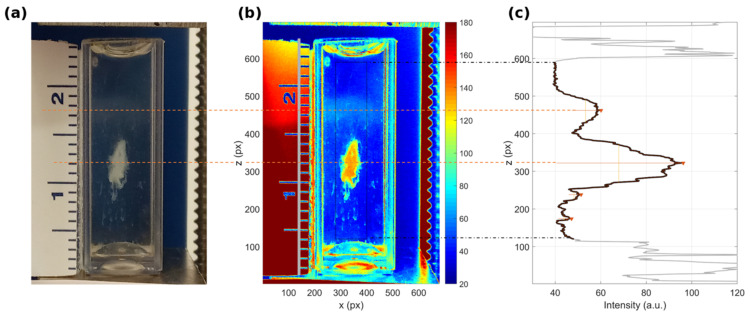
Representative outcome of the image processing. (**a**) Optical image of a MagLev pattern. (**b**) Corresponding pseudo-color image. (**c**) Evaluation of the intensity profile and determination of locations, amplitude, and width of the detected peaks. Dashed lines indicate the vertical limits of the region of interest (black) and the computed location of the detected peaks (red).

**Figure 3 cancers-13-05155-f003:**
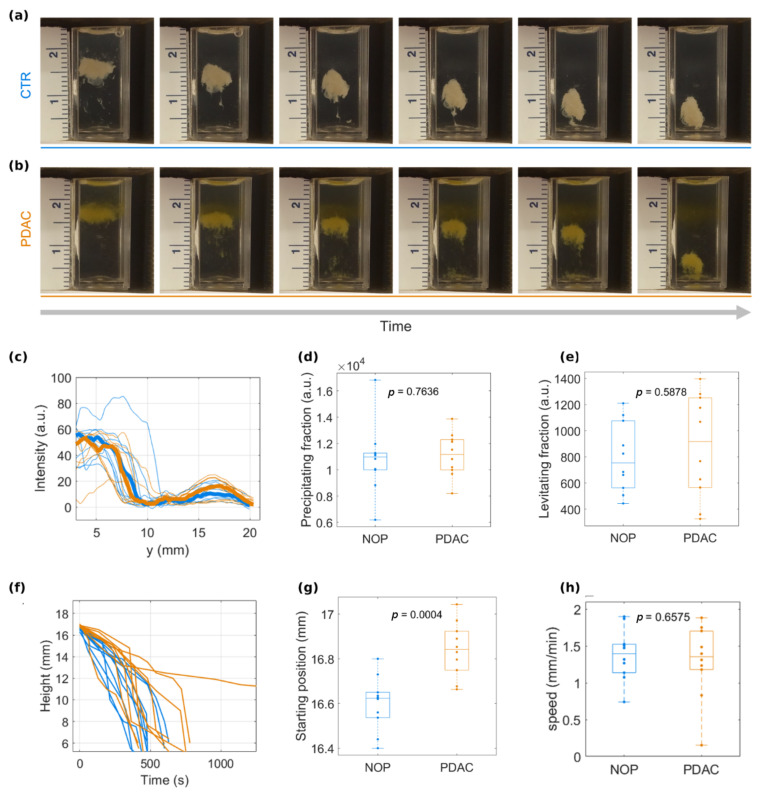
MagLev patterns of GO-PC for (**a**) non-oncological and (**b**) PDAC samples. (**c**) Intensity profiles for the 20 investigated samples (averages are reported as thick lines) at the maximum time point, and corresponding distributions of integral areas for (**d**) precipitating and (**e**) levitating populations. (**f**) Time evolution of the precipitating components and corresponding distributions of (**g**) peak’s starting position and (**h**) peak speed. *p*-value from Student’s *t*-test are reported in panels (**d**,**e**,**g**,**h**).

**Figure 4 cancers-13-05155-f004:**
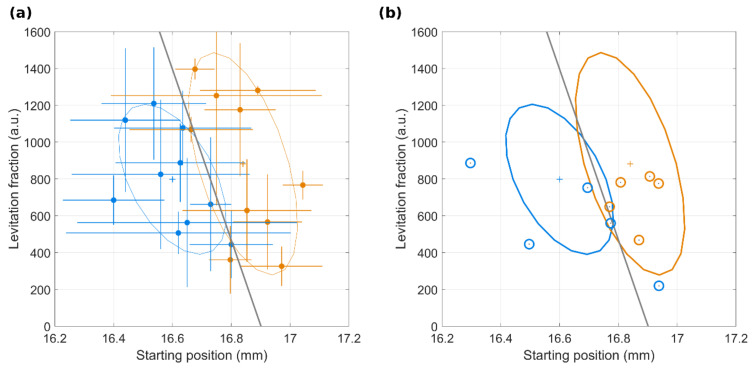
(**a**) Scatter plot of the MagLev pattern’s starting position and levitating population area for a training set of N_C_ = 10 NOP samples (blue dots) and N_P_ = 10 PDAC samples (orange dots) and corresponding experimental errors. Crosses indicate the average positions, ellipses the distribution extensions and the black line is the output of a linear discriminant analysis (LDA). By LDA computation, the resulting test’s parameters read specificity = 100%, sensitivity = 80%, global accuracy 90%. (**b**) Output of a validation blind test superimposed to the distributions of the training dataset (ellipses) and the outcome of the linear discriminant analysis (black line). N_C_ = 5 NOP samples (blue circles), N_p_ = 5 PDAC sample (orange circles).

## Data Availability

The data presented in this study are available on request from the corresponding author.

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
