# Peer review of "Detection of Pancreatic Ductal Adenocarcinoma by Ex Vivo Magnetic Levitation of Plasma Protein-Coated Nanoparticles"

_cancers, 2021, doi:10.3390/cancers13205155_

Round 1

Reviewer 1 Report

I commend the authors for the tremendous amount of work that they did so quickly. The manuscript is much improved and I only have a few minor comments about organization. After these have been addressed the manuscript can stay with the editor (I do not feel that it needs to be sent back to me for acceptance).

To a reader, I believe Figure S4a is far more convincing than Figure 4a. It really demonstrates that the data clusters into two distinct groups (and is exactly the same plot). I would strongly recommend replacing Figure 4a with Figure S4a. Figure S4b should stay supplemental because it just demonstrates reproducibility between different people.

Minor Points:

(1) I believe the term “Accuracy” is more appropriate than “Correctness” because it is strictly defined as (TP + TN)/(TP + TN + FP + FN). Correctness doesn’t really have a formal definition. The authors have an Accuracy of 90% so the word Correctness should just be replaced with Accuracy throughout the manuscript.

(2) The way the patient demographics are presented is very atypical.

Table S1 and S2 should be combined. The only reason to have these tables is to compare the two patient populations and it is impossible to do so when they are on top of each other. They should be side by side, preferably with p values demonstrating which demographics are different and which are NS.

This should be done for Table S3 -Table S4 & Table S1b-Table S2b & Table S3b-S4b (but without a p-value calculated).

I don’t think the male/female part for each comorbidity is necessary in Table S3, S4, s3b, s4b because it doesn’t matter if it is a hypertensive man or hypertensive woman – the comorbidity is hypertension.

The nomenclature of Table S1b, S2b, S3b, S4b is confusing.

To correct all of these points I would recommend:

New Table S1 = Table S1 + Table S2 w/ P-values

New Table S2 = Table S3 + Table S4 (no P-values because very low incidence, no Male / Female)

New Table S3 = Table S1b + Table S2b (? P-values because only 5 people/group)

New Table S4 = Table S3b + Table S4b (no P-values, no Male / Female)

(3) Line 393-5. NOP and PDAC are mixed up. The authors wrote “Such higher protein concentration may result in faster precipitation thus accounting, at least in part, for the lower starting position observed for PDAC (Fig. 3g).” I believe they mean to say lower starting position observed for NOP because these patients have the higher protein concentration that will settle faster.

(4) Line 299 – Inter, not Inte-user

(5) Lines 414-5. In addition to obesity& DMII, smoking should be added as a risk factor for PDAC.

Author Response

All the points raised by the Reviewer have been addressed by the authors.

Reviewer 2 Report

The authors addressed points raised by reviewers and revised properly.

Author Response

All the points raised by the Reviewer have been addressed by the authors.

This manuscript is a resubmission of an earlier submission. The following is a list of the peer review reports and author responses from that submission.

Round 1

Reviewer 1 Report

In this manuscript, Digiacomo et al. propose using a new technology MagLev applied to patient’s serum to aid in the diagnosis of pancreatic ductal adenocarcinoma. This study is intriguing; however, as the manuscript currently stands the data appears comes off more like a preliminary pilot than a flushed out clinical/translational study.

Major comments

1) Lines 253-4. The methods say that “after a 1hr incubation at 37C, each of the samples was pipetted into a dyspropium solution and the system was inserted into the MagLev device.” Since the ‘starting point’ was one of the two significant variables they used in their final analysis can this reviewer assume that this step was NOT performed manually. The depth and angle of insertion of the pipet tip as well as the viscosity, density, speed of injection, and agitation could all certainly affect the starting point to a degree greater than the reported 1 mm difference (Fig 3g). To address this concern, can the authors please provide a detailed method section including experimental equipment as to how this important step was performed. This is of course unless the solutions were mixed and vortexed to homogeneity; however, I assume this was not done because otherwise the precipitate would be homogeneously distributed throughout the vial like a snow globe. The authors also discuss how this technique is cheap and easily performed – if two different people inject the same sample is the starting point identical?

2) Does the ‘starting point’ refer to the low density, the high-density proteins, or a combination of the two. I assume it is the combination of the two because it was not explicitly stated. Since the vast majority of the signal at the start is from the high-density precipitate, this variable appears to speak mostly to everything that is NOT bound to graphene because it doesn’t levitate and just falls out of solution. This is confusing because their final variable is then generated from data about the material that levitates combined with the proteins that precipitate. In what fraction do the oncologic proteins reside or do both fractions contain the oncologic proteins?

3) Since the technique of MagLev has never been applied to patient serum as the authors state in line ~334, there is no information on the technical reproducibility of this technique run multiple times on a single sample. This is especially important with respect to comment 4 below. The authors should run several samples in triplicate to demonstrate the standard deviation of each of the variables discussed. Especially, the starting point if this was done manually.

4) There do seem to be two distinct groups; however, as the manuscript current stands it is very difficult to interpret the cut-off line. To use an American term, the cut off in Figure 4A appears to be Gerrymandered. Since none of the samples were run in duplicate or triplicate and several lie close to the pos/neg cutoff it is very hard to comment on the robustness of this line. A completely independent validation set is likely required. Further, since the starting point (x-axis) is largely reporting on the precipitate (high density material) and the y-axis refers to the low-density levitating fraction, this reviewer is curious why these form a composite cut-off since the oncologic proteins would theoretically be in only one of these fractions.

5) How were the patient’s chosen? Were these consecutive patients and thus no patient samples were excluded. E.g. no selection bias.

6) What were the concentrations of total protein, albumin, and albumin free protein values for the 20 sera used. This is important because sedimentation / precipitation of the high-density group will depend on aggregation size. Generally, the higher the protein concentration the larger the particulate size and the faster they will settle. Further, to make an analogy between the passive adsorption to graphene and other passive adsorption processes (e.g. ELISA) blocking with native proteins plays an important role in specificity and signal. Thus, if there was less protein (or albumin) present in patients with advanced cancer (80% of patients were stage III or greater) this could be an important variable to explain their results. In other words, with low serum protein a different population of proteins might be able to bind the graphene. Lastly, if the serum from the normal patients had higher protein resulting in quicker sedimentation could the high density proteins pull the graphene bound low density proteins to the bottom and this is why there is less low density area in NOP.

7) Lines 314-6, referring to one’s own work as “seminal” comes with significant hubris, I would recommend rewording this sentence without that term.

8) Were these the same 20 patient samples used in Di Santo et al. 2020.

9) Figure 3 – data points should be shown over the Box and Whisker plots

Minor Comments:

10) The authors reported ratios of cells for SIRS biomarkers – what about traditional biomarkers or tests like CRP or ESR for inflammation. What was the CA19-9 levels (and CEA if measured) on these patients – did that correlate with distance from the cutoff in Figure 4A.

11) What happens to the low- and high-density fractions when the amount of serum added to the reaction is reduced?

12) Line 45 is very speculative. One paper suggests this. Considering deleting this – everyone reading this manuscript will know early detection of PDAC is important.

13) In addition to proteomics consider briefly discussing glycomics as CA19-9 is a glycosylation epitope detected with antibodies and Das-1, which has much higher accuracy at least for IPMN analysis has also recently been found to bind a glycosylation epitope.

14) Are the authors sure that only proteins precipitate in the dysprosium solution: could these be lipids, carbohydrates, small molecules, etc?

15) Are the low-density floating samples really of lower density than the high density precipitation or does that fraction float because of the neodymium magnets. In which case maybe it would be better to call them the levitating fraction and the precipitating fraction.

16) Much of the introduction and discussion focuses on early detection of PDAC. I agree this is important; however, the patient samples reported here do not reflect this, 80% of the patients were stage III or higher. Maybe the authors could tone down the early detection notes and also discuss how hard it is to discriminate pancreatic cancer from the parenchyma using contrasted CT, EUS, or MRI, which also contributes to late diagnosis. And say alternative, complementary tests could aid in the diagnosis and play a role in surveillance intervals.

Reviewer 2 Report

This topic is very comprehensive and informative, and there is almost nothing that needs to be corrected. The following modifications are required to improve the manuscript:

Major comments

  1. The introduction part details the background of the study, but it is too long. It would be more appealing to readers to summarize and describe more clearly the issues that this study addresses. And the necessary and detailed description should be written in the discussion part.
  2. To improve the readability of the introduction and discussion, please add line breaks. If the lack of line breaks is due to the manuscript for a peer reveiw, then please delete this comment.
  3. From the results of the study, it can be speculated that large and high density molecules are lost in the plasma of PDAC patients compared to that of healthy individuals. I would like you to add a consideration of the lost populations to the discussion.
  4. Figure 4. I would like you to add validation cohort to confirm the differentiation ability of this statistical analysis, if possible. It is preferable to include Stage I and II in the validation cohort.

Minor comments

Page 7, line30. This section (4.2.) includes the contents about the anticancer activity of CPPs. This subsection (4.2.4) does not include anticancer activity but the cellular entry mechanism of arginine amino acids, unlike the above three subsections (4.2.1/4.2.2/4.2.3). Please consider moving the content of this subsection to a different part and reorganize the text. 
